# Ultraviolet Light Effects on Cobalt–Thiourea Complexes Crystallization

Luis Eduardo Trujillo Villanueva [1] , Felipe Legorreta García [1], Fidel Pérez Moreno [1], Marius Ramírez Cardona [1] and Edgar Arturo Chávez Urbiola [2],*

[1] Área Académica de Ciencias de la Tierra y Materiales, Instituto de Ciencias Básicas e Ingenierias, Universidad Autónoma del Estado de Hidalgo, Pachuca de Soto 42184, Hidalgo, Mexico; tr162855@uaeh.edu.mx (L.E.T.V.); profe_974@uaeh.edu.mx (F.L.G.); fpmoreno@uaeh.edu.mx (F.P.M.); mariusr@uaeh.edu.mx (M.R.C.)

[2] Consejo Nacional de Ciencia y Tecnología, Área Académica de Ciencias de la Tierra y Materiales, Instituto de Ciencias Básicas e Ingenierias, Universidad Autónoma del Estado de Hidalgo, Pachuca de Soto 42184, Hidalgo, Mexico

\* Correspondence: edgar_chavez@uaeh.edu.mx

**Abstract:** In this work, a cobalt–thiourea complexes crystal synthesis was carried out comparatively with and without ultraviolet light assistance ($\lambda$ = 253 nm), and its effect was studied. Through the solvent evaporation technique, crystalline forms were obtained, which were analyzed and characterized by different techniques: Raman spectroscopy, X-ray diffraction (XRD), and digital optical microscopy. Crystal's shape changes were observed when comparing those obtained from the solution with and without ultraviolet (UV) assistance. It was found that the UV light effect on the crystals causes a structural modification of the complex synthesized in the (022) (120) planes and without UV assistance in the (002), (111), (13$\bar{1}$), and (13$\bar{2}$) planes. It is also possible to observe an increase in intensity by Raman spectra identified as Co–S bonds (297 cm$^{-1}$) for crystals synthesized with UV assistance.

**Keywords:** crystal growth process; UV light assistance; cobalt–thiourea complex; thiourea compounds; crystal morphology

## 1. Introduction

The combination of metals (M) located in group B of the periodic table plus thiourea ($SC(NH_2)_2$) is often used for crystal growth, thereby achieving the formation of thiourea complexes ($[SC(NH_2)_2]_x + M$) [1]. The thiourea complex used for this investigation is called (bis) thiourea cobalt chloride $[CS(NH_2)_2]_2 \cdot CoCl_2$, which can be formed from cobalt chloride hexahydrate $CoCl_2 + 6H_2O$ plus thiourea $SC(NH_2)_2$ [2], as shown in the following reaction:

$$CoCl_2 + 2[CS(NH_2)_2] \rightarrow [SC(NH_2)_2]_2 \cdot CoCl_2 \tag{1}$$

For crystallization, the solvent evaporation technique was used. Solvent evaporation was carried out under controlled conditions for each of the experiments [3]. Investigate the effects of ultraviolet light on cobalt–thiourea complexes crystals is the main motivation of this investigation. For this, electromagnetic radiation's assistance will be used in a specific case, ultraviolet light (wavelengths $\lambda$ lower than 400 nm), to provide the energy that allows the crystal's physical modification.

Ultraviolet light has been used to synthesize films of ZnS [4], CdS [5]. These authors reported a photochemical bath deposition (PCBD) technique in which the sulfate precursor solution absorbs light with a wavelength shorter than 300 nm. Some it is because of this that a more energetic ultraviolet radiation (wavelength less than 400 nm) is applied, which promotes reactions in the solution and the fix of deposited films. On the other hand, using ultraviolet light focused directly on crystal growth has been studied in recent years. As

an example, we have the di-thiourea cadmium chloride crystals complex assisted with ultraviolet light, which it is mentioned that the crystals exposed to UV light present a preferential growth compared to the crystals without UV assistance [6]. It is also important to mention that some of these thiourea complexes have non-linear optical properties. Such compounds have been studied in recent years [7,8].

## 2. Materials and Methods

Cobalt chloride hexahydrate (Alyt, 98% purity) and thiourea (Reasol®, technical grade) were used for synthesizing (bis) thiourea cobalt chloride. The experimentation was carried out in an acid medium. Hydrochloric acid (JT Baker at 36.5% concentration) was used in a ratio of 0.2 mL/250 mL of ethanol to obtain a solution with a pH of 2.0.

### 2.1. Characterization

The characterization was carried out by X-ray diffraction (XRD) using an Inel device with a cobalt source (Equinox 2000, Artenay, France). For the diffuse reflectance spectrophotometry characterization, an Ocean optics model DT 1000 CE UV-vis spectrometer made in United States of America was used. Then, for the Raman dispersion characterization, a LabRAM HR Evolution Jobin Yvon Technology made in France was used. Finally, a digital microscope (interpolated to 2.0 MPIX) model Celestron 44302-C made in Torrance (CA, USA), United States of America, purchased on the official website was used to analyze the size of the crystals.

### 2.2. (Bis) Thiourea Cobalt Chloride Synthesis

The synthesis of the cobalt–thiourea complex was performed by the solvent evaporation technique (SET). To achieve the synthesis, 2.37 g of cobalt chloride hexahydrate and 1.52 g of thiourea were poured in 10 mL of ethanol at pH 2. The solution was kept under constant stirring and at a constant temperature of 60 °C for 10 min, achieving a complete dissolution. Afterward, the solution was filtered and poured into a 60 mm glass Petri dish. The uncovered glass Petri dish with the solution was placed inside a container with ultraviolet light sources on top for ultraviolet assistance. A representative scheme of the process is shown in Figure 1. The ultraviolet light source was kept on for 3 h. At the end of the time, the glass Petri dish was covered and placed in an isolated chamber with a controlled temperature of 25 °C for 12 days to allow the crystallization process.

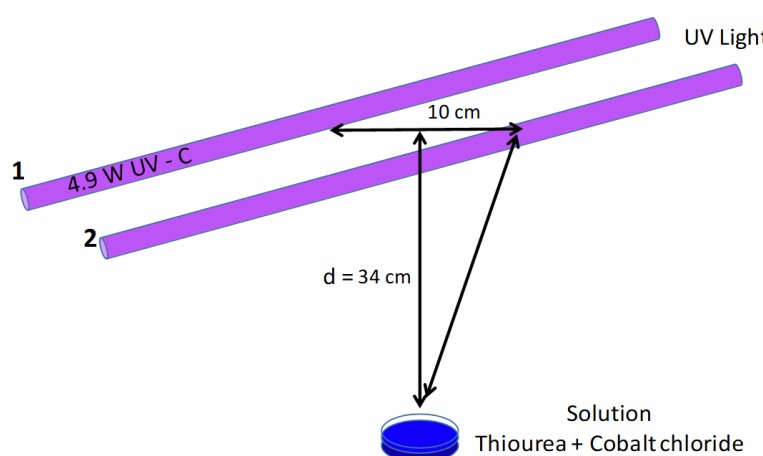

**Figure 1.** Experimental scheme of the distance between the UV source and the solution.

In the second experiment, the same amounts as in the first experiment were used: cobalt chloride hexahydrate (2.37 g) and thiourea (1.52 g) in 10 mL of ethanol at pH 2. Again, under constant stirring and at a constant temperature of 60 °C for 10 min, until complete dissolution. In the end, the solution was filtered and poured into a 60 mm glass Petri dish.

The glass Petri dish with the solution was placed inside a container without ultraviolet light sources for 3 h. The glass Petri dish was covered and placed in the same isolated chamber used in the first experiment for 12 days. In both experiments, crystallization started after 3 days, and the solvent was evaporated entirely after 12 days.

The UV source observed in the representative scheme of Figure 1 corresponds to two lamps commonly used for sanitizing, distributed by SaniLIGHT®as UV lamp model D36-2s 40-0178C made in United States of America, UV source is a STER-L-RAY lamp model G36T5L manufactured by Atlanta Ultraviolet, made in New York, NY, United States of America. The lamp power mentioned above in the total output is 15 W and has a radius of 15 mm and a length of 84.3 cm. Therefore, the total radiation emitted by the two UV source would be 27.6 W/m$^2$. These data were obtained using geometric calculations based on the performance reported by the manufacturer.

Using an Ocean optics model DT 1000 CE UV-vis spectrometer, the emission ranges of the UV source were obtained, registering a greater intensity in the wavelength at 253 nm, as shown in Figure 2.

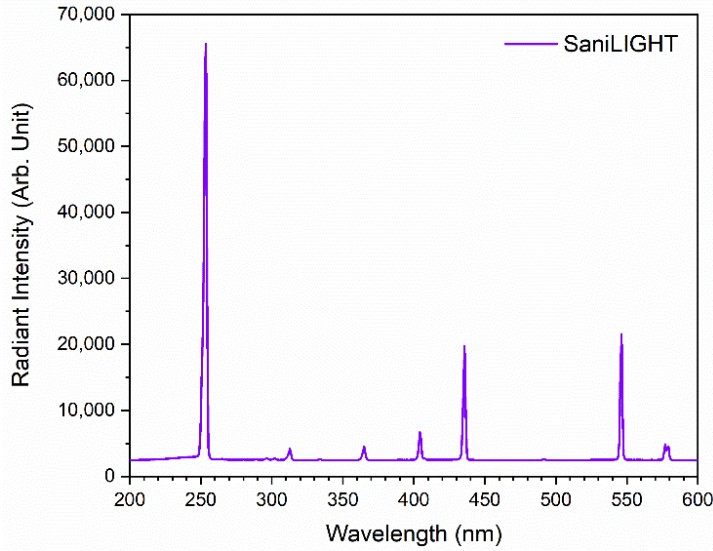

**Figure 2.** The emission spectrum of the light source used (SaniLIGHT®).

### 3. Results

The absorption spectrum of the precursors was studied individually. For this, 0.1 molar solutions were prepared with the reagents used in the experimentation, each in an ethanol solution. This radiation was measured using a UV-Vis spectrometer at an integration time of 30 ms and averaging 10 measurements in quartz cells. Figure 3a exhibits the thiourea solution spectrum, which has an absorption range of 225 to 270 nm. The cobalt chloride hexahydrate spectrum, Figure 3b, shows an absorption range between 200 and 250 nm. Finally, the combination of both reagents spectrum, Figure 3c, shows similar behavior to that obtained in thiourea. An absorption range from 225 to 270 nm is observed with maximum absorption in the wavelength of 243 nm. It is important to note that ultraviolet assistance occurs in the solution. For this reason, knowing the absorption range of the precursor solution allows the correct choice of the wavelength of the emitting source.

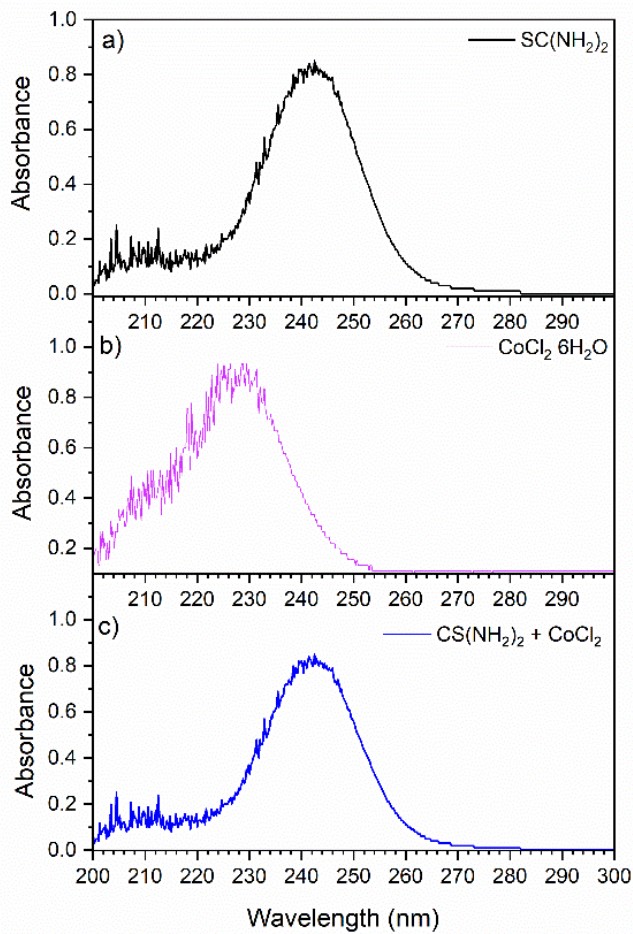

**Figure 3.** Absorption ranges of solutions: (**a**) SC(NH$_2$)$_2$; (**b**) CoCl2 · 6H2O; (**c**) [SC(NH$_2$)$_2$]$_2$ · CoCl$_2$.

The obtained crystals without UV radiation are shown in Figure 4a, while the UV-assisted crystals are shown in Figure 4b. Different sizes of crystals are observed; those assisted by UV light show a more significant growth size. A change in the geometry of the crystals is noticeable; a defined geometric shape, hexagonal and others in a rhombohedral shape. On the other hand, crystals without UV assistance show a smaller growth, with an average size of 0.2 cm. The size of some crystals obtained with ultraviolet light shows a larger size compared to crystals obtained without ultraviolet light.

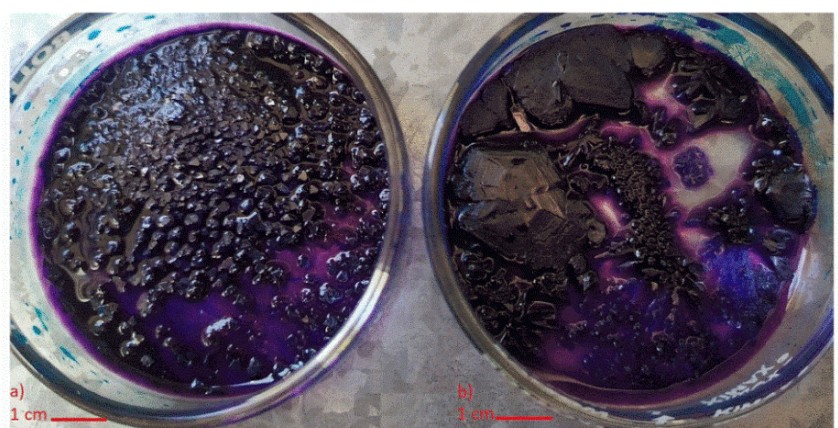

**Figure 4.** (**a**) crystals without UV assistance; (**b**) crystals with UV assistance.

### 3.1. Optical Microscopy

The crystals' morphology details were characterized by digital microscopy with a digital microscope (interpolated to 2.0 MPIX) model Celestron 44302-C made in California, United States of America, as shown in Figure 5, which shows the crystals obtained without UV assistance, agglomerated crystals were observed. Most of them do not present a defined shape, but some structures similar to a trapezoid stand out, which can be observed in some agglomeration points.

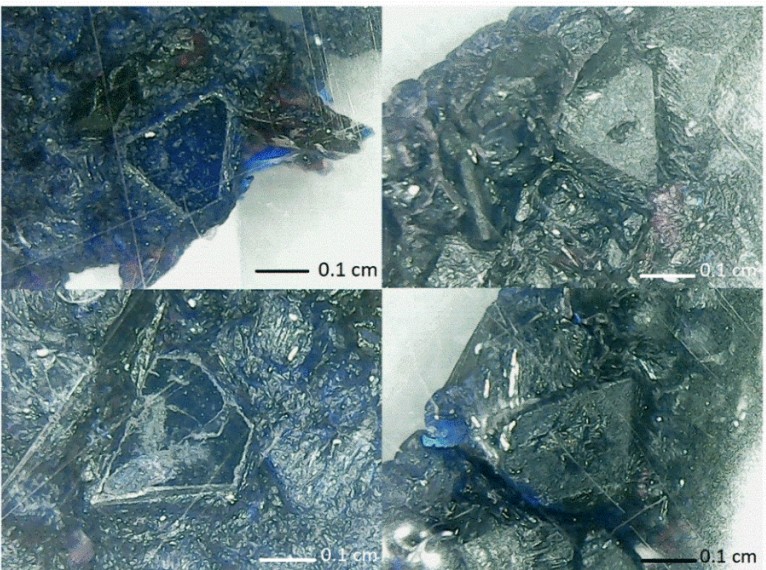

**Figure 5.** View of crystals without UV assistance by digital microscopy.

The following statistical analysis (Figure 6) shows the average size of the crystals without UV.

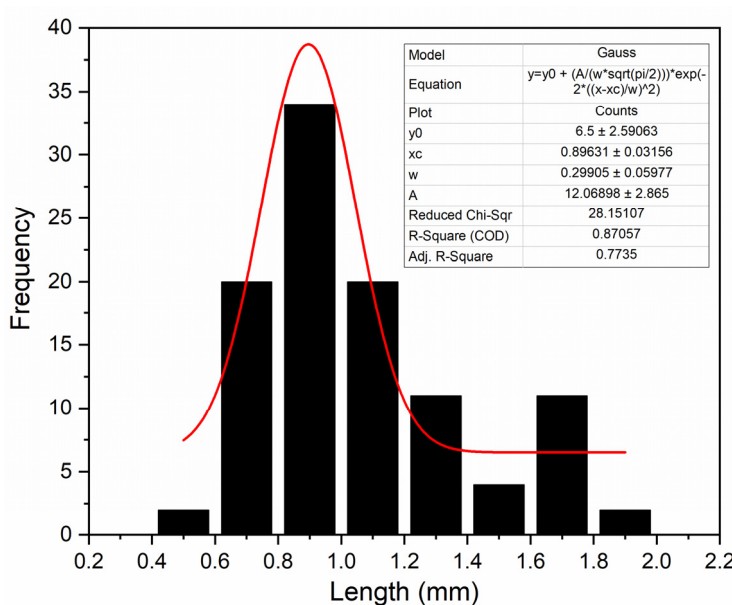

| Model | Gauss |
|---|---|
| Equation | y=y0 + (A/(w*sqrt(pi/2)))*exp(-2*((x-xc)/w)^2) |
| Plot | Counts |
| y0 | 6.5 ± 2.59063 |
| xc | 0.89631 ± 0.03156 |
| w | 0.29905 ± 0.05977 |
| A | 12.06898 ± 2.865 |
| Reduced Chi-Sqr | 28.15107 |
| R-Square (COD) | 0.87057 |
| Adj. R-Square | 0.7735 |

**Figure 6.** Statistical analysis of crystals without UV assistance.

There is an average size of 0.9 mm in 32.2% of the crystals:
Average size ($xc$) = 0.9 mm
Peak width ($w$) = 0.29 mm

Standard deviation:

$$(\sigma) = w/2 = 0.145 \text{ mm}$$

Therefore, average size with error = $xc \pm \sigma$:

$$0.9 \pm 0.29 \text{ mm}$$

Polydispersity $(\sigma/xc) \times 100\% = (0.29/0.9) \times 100\% = 32.2\%$.

In contrast, the UV light source exposed crystals exhibit an average size of 0.4 cm, as is shown in Figure 7. Those crystals show more defined geometric shapes, hexagonal and rhombic. The shapes obtained are similar to those reported by the SET technique but after 20 days [9].

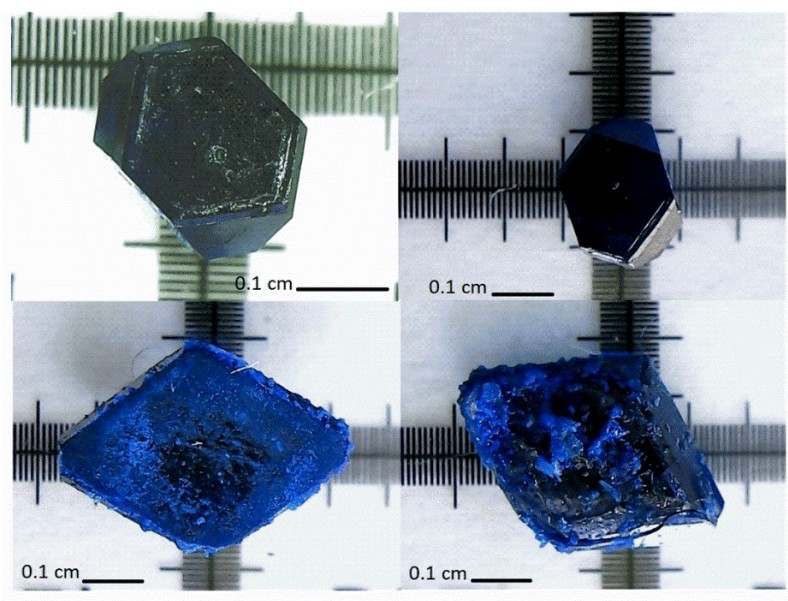

**Figure 7.** View of crystals with UV assistance by digital microscopy.

The following statistical analysis (Figure 8) shows the average size of the crystals grown with UV light influence.

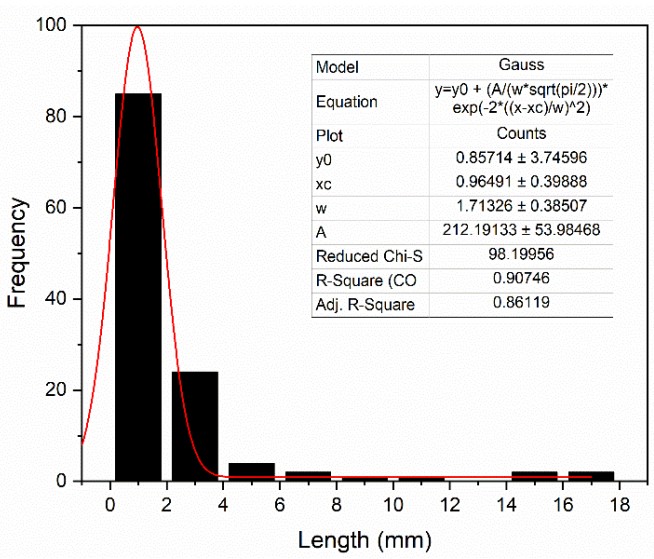

| Model | Gauss |
|---|---|
| Equation | y=y0 + (A/(w*sqrt(pi/2)))* exp(-2*((x-xc)/w)^2) |
| Plot | Counts |
| y0 | 0.85714 ± 3.74596 |
| xc | 0.96491 ± 0.39888 |
| w | 1.71326 ± 0.38507 |
| A | 212.19133 ± 53.98468 |
| Reduced Chi-S | 98.19956 |
| R-Square (CO | 0.90746 |
| Adj. R-Square | 0.86119 |

**Figure 8.** Statistical analysis of crystals with UV assistance.

Average size $(xc)$ = 0.96 mm

Peak width ($w$) = 1.71 mm
Standard deviation:

$$(\sigma) = w/2 = 0.85 \text{ mm}$$

Therefore, average size with error = $xc \pm \sigma$:

$$0.96 \pm 0.85 \text{ mm}$$

Polydispersity $(\sigma/xc) \times 100\% = (0.85/0.96) \times 100\% = 88\%$.

### 3.2. X-Ray Diffraction

The X-ray diffraction characterization was performed with an Inel device (Equinox 2000) using a cobalt source. The first pattern is shown in Figure 9; in section (a), the crystals obtained XRD pattern without UV assistance is located. In contrast, in section (b), the pattern was obtained from the crystals with UV assistance. In indexing to section (a), it coincides with the powder diffraction pattern (PDF 00-052-0587) corresponding to the (bis) thiourea cobalt chloride $[SC(NH_2)_2]_2 \cdot CoCl_2$ [10].

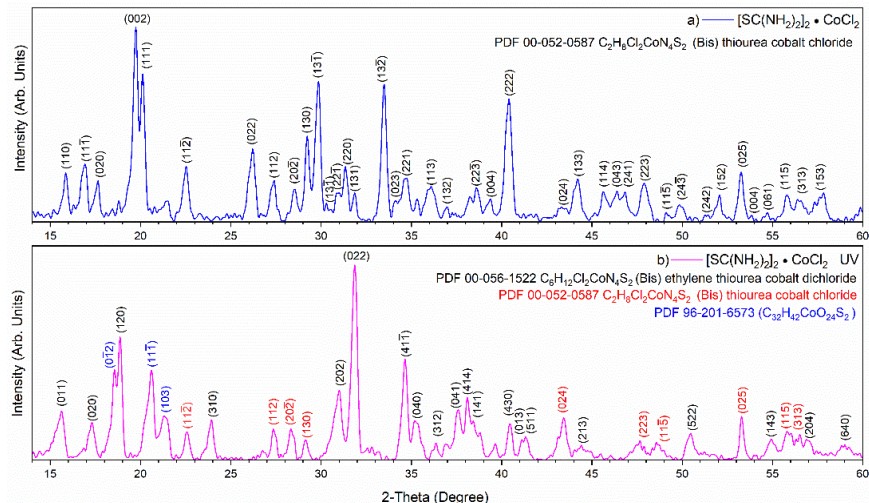

**Figure 9.** XRD pattern: (**a**) crystals without UV assistance; (**b**) crystals with UV assisted.

In section (b), with UV assistance, different crystalline phases are shown; first, black color numbered planes coincide with the powder diffraction pattern (PDF 00-056-1522), which corresponds to (bis) ethylene thiourea cobalt dichloride [11]. The red color numbered planes coincide with (PDF 00-052-0587), which corresponds to the (bis) thiourea cobalt chloride showed in section (a). The $(0\bar{1}2)$, $(11\bar{1})$ and (103) blue planes, match the powder diffraction pattern (PDF 96-201-6573, $C_{32}H_{42}CoO_{24}S_2$) [12], another thiourea complex. Both diffraction patterns retain some similarities to each other when comparing the intensities of the planes in red color for (bis) thiourea cobalt chloride, but other complexes are also obtained (planes in black and blue). The pattern obtained from the crystals without ultraviolet assistance shows a monoclinic structure called (bis) thiourea cobalt chloride in its entirety. In contrast, the diffraction pattern obtained from the crystals with ultraviolet assistance has a polycrystalline composition. The structure called (bis) ethylene thiourea cobalt dichloride is a monoclinic structure, which is incorporating part of the solvent used (ethanol) in its crystalline structure, in addition to forming other complexes, such as $C_{32}H_{42}CoO_{24}S_2$ and partly (bis) thiourea cobalt chloride. Although the UV-assisted crystal structure has a more defined crystalline geometry, other cobalt–thiourea complex forms are also obtained.

### 3.3. Raman Spectroscopy

The obtained crystals were characterized by the Raman technique; these results are shown in Figure 10. The crystals synthesized without UV assistance are shown in section (a) and with UV assistance in section (b). Both crystals show the same bonds but with different intensities. It can be seen that the crystals with UV assistance present a higher intensity for the Co–S bond (297 cm$^{-1}$). Increasing the Co–S bonds (297 cm$^{-1}$) decreases the Co–Cl bonds to 255, 237, and 263 cm$^{-1}$. In contrast, for the crystals synthesized without UV assistance, the highest intensity corresponds to the Co–Cl bond (263 cm$^{-1}$). Different references were used to identify the Raman study signals [13–15]. The increase in intensity in the wavelength at 297 cm$^{-1}$ in crystals exposed to UV light has already been reported in other thiourea complexes, such as di-thiourea cadmium chloride crystals [6]. It is possible to observe an increase in the Co–S bonds (297 cm$^{-1}$), which suggests that ultraviolet light favors the incorporation of thiourea, specifically sulfur ions, to cobalt metal ions. To complement this information, Table 1 shows the identification of the vibration modes for each of the bonds found in both crystals.

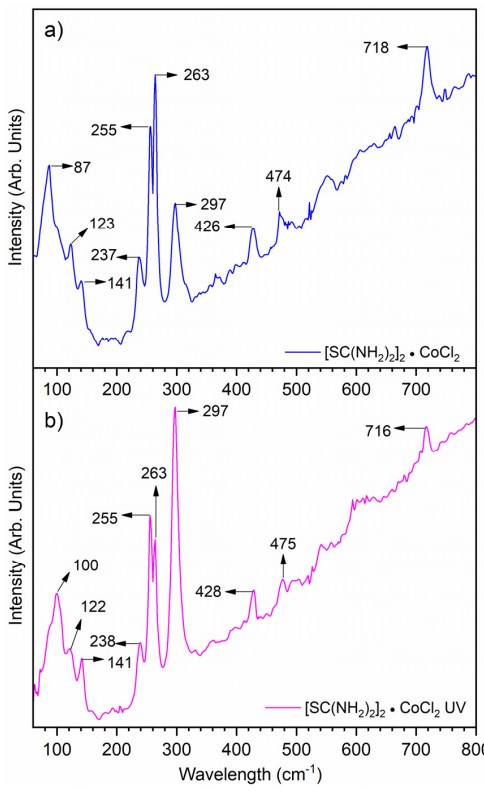

**Figure 10.** [SC(NH$_2$)$_2$]$_2$ · CoCl$_2$ Raman spectra (**a**) without UV assistance; (**b**) with UV assistance.

**Table 1.** Vibrational modes in cobalt–thiourea complex crystals without and with UV assistance.

|  | [CS(NH$_2$)$_2$]$_2$ · CoCl$_2$ without UV Assistance (cm$^{-1}$) | [CS(NH$_2$)$_2$]$_2$ · CoCl$_2$ with UV Assistance (cm$^{-1}$) |
|---|---|---|
| (C–S)$_w$ | 718 | 716 |
| (N–C–S)$_w$ | 474 | 475 |
| (H–N–H)$_w$ | 426 | 428 |
| (Co–S)$_{vs}$ | 297 | 297 |
| (Co–Cl)$_{vs}$ | 263 | 263 |
| (Co–Cl)$_s$ | 255 | 255 |
| (Co–Cl)$_w$ | 237 | 238 |
| (Co–S-C)$_w$ | 141 | 141 |
| (Co–Cl)$_w$ | 123 | 122 |
| (Co–Cl)$_m$ | 87 | 100 |

vs: very strong; s: strong; m: medium; w: weak.

## 4. Discussion

The experimental conditions in both cases are the same; the only difference is that one solution is irradiated with ultraviolet light; it leads to a variation that modified the final crystallization result. The results obtained in this study provide evidence that the ultraviolet light assistance has effects on bis thiourea cobalt chloride $[SC(NH_2)_2]_2 \cdot CoCl_2$ modifying its crystalline structure. The observed effects in the morphology are quite evident since the crystal increases its growth rate reaching a size up to 10 times larger than the crystal obtained without UV assistance. On the other hand, the crystals obtained from the solution exposed to the UV light source showed more defined geometric shapes, hexagonal and rhombic shapes.

XRD exhibits crystalline structure changes caused by the UV light, showing a structural modification of the (bis) thiourea cobalt chloride mainly in the $(002)$, $(111)$, $(13\bar{1})$, $(13\bar{2})$ and $(222)$ planes.

Raman results show an intensity increase of the bonds associated with Co–S in the crystals assisted by UV light. Therefore, if there is an increase in Co–S bonds, there will be a decrease in Co–Cl bonds, which favors the modification in the geometric shape of the crystals. In addition, it is evident that to achieve the modification of the crystals. The ultraviolet light source must be within the absorption range of the solution used to grow crystals. Otherwise, it will not affect the thiourea complex.

When identifying the absorption ranges, the thiourea has a range close to the emission range of the ultraviolet source. Therefore, the interaction of the UV light with the solution will occur directly in the thiourea, providing additional energy, which will favor the incorporation of thiourea over cobalt chloride. Consequently, as there is a greater quantity of thiourea bound to cobalt, the Co–Cl bonds (255, 237, and 263 cm$^{-1}$) will decrease.

In this same research line, similar results were obtained in another complex, such as (bis) thiourea cadmium chloride, inducing a preferential growth by the assistance of ultraviolet light. Such an effect has been observed in other thiourea complexes, which, when irradiated with ultraviolet light, favors the union of thiourea to the metal used in the chemical reaction [16]. It is important to mention that the ultraviolet light source in the UV-C range (253 nm) produces ozone inside the container; the ozone formed must be extracted as it prevents the formation of thiourea complexes.

**Author Contributions:** Conceptualization, L.E.T.V. and E.A.C.U.; methodology, L.E.T.V. and E.A.C.U.; software, M.R.C.; validation, F.L.G., F.P.M. and M.R.C.; formal analysis, L.E.T.V.; investigation, L.E.T.V.; resources, F.L.G. and E.A.C.U.; data curation, M.R.C. and F.P.M.; writing—original draft preparation, L.E.T.V.; writing—review and editing, L.E.T.V., E.A.C.U., F.L.G.; visualization, L.E.T.V.; supervision, E.A.C.U.; project administration, E.A.C.U.; funding acquisition, F.L.G., E.A.C.U., M.R.C., F.P.M. All authors have read and agreed to the published version of the manuscript.

**Funding:** This research received no external funding.

**Acknowledgments:** We acknowledge the technical assistance of Ing. Francisco Rodríguez Melgarejo and Sergio J. Jiménez Sandoval from the Raman spectroscopy lab in CINVESTAV Qro. In addition, for the "Catedras CONACyT" program, project 1226. In addition, for the CONACyT Graduate Scholarship of L. E. Trujillo. Finally, to the UAEH for the facilities.

**Conflicts of Interest:** The authors declare no conflict of interest.

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
