# Peer review of "Ultraviolet Light Effects on Cobalt–Thiourea Complexes Crystallization"

_crystals, doi:10.3390/cryst11050473_

Round 1

Reviewer 1 Report

I am against publishing this manuscript. Please read comments below:

The authors describe “a method” of preparation of big size crystals of CoCl2(SCN2H4)2 but they do not control basic parameters and the conclusions drawn do not explain many aspects of the searched system which seems to be out of their control. Actually we are not sure about the composition of the final crystals in both experiments. Is there a chemical reaction with ethanol that produces stated bis(ethylenethiourea) cobalt dichloride? It would not be just a modification of crystal growth of a defined substance.   

No experiment showing NLO property of the UV-assisted product was performed.

Obviously Co-Cl bond lengths will be different in coordination compound with two thiourea ligands and four thiourea ligands and two or four ethylenethiourea ligands (ca 2.468 A in tetragonal structute CoCl2(tu)4 Murugan 2008, and ca 2.26 A in monoclinic CoCl2(tu)2 structures).  

Cited references 2 and 4 are inconsistent since they both describe tetragonal (space group P42/n) bis chlorido tetrathiourea cobalt(II), the same unit cells are given, despite the title of the reference 2 (where no atom coordinates were given) which says about bis chlorido bisthiourea cobalt(II).

The authors also do not cite determination of real dichlorido bisthiourea cobalt(II) structures in monoclinic groupc Cc (P.Domiano, A.Tiripicchio, Cryst.Struct.Commun. (1972), 1, 107;   and more recently: Shefali Vaidya, S.K.Singh, P.Shukla, K.Ansari, G.Rajaraman, M.Shanmugam, Chem.-Eur.J. (2017), 23, 9546) or its polymorph in P21/n group (M.Dhandapani, M.A.Kandaswamy, M.Nethaji, CSD Communication (2016)). Due to symmetry (presence of inversion centers) the last one cannot exhibit  NLO properties.

Minor issues

Text in lines 55-61 is repeated exactly in lines 66-71 without alteration. This is in bad style and hinders the main idea.

Line 81 (check units and proportion) states the lamp radius as 35.4 cm. Should it not be 35.4 mm?

Author Response

Our research team appreciates the time spent reviewing this article. In an attached file in Word format will be the answer to each comment, again we appreciate your comments.

Reviewer 2 Report

The article Entitled "Ultraviolet light Effects on cobalt-thiourea complexes crystallization" deals with difference of characteristic of crystal of thiourea cobalt crystal when one was exposed to UV irradiation and the other no. 

The article is well written. first they put the bakcground and he purpose of the article, secondly they explained their methods of UV-assisted crystallization and after the different analysis exposed, they conclude. 

The article need several modification to be publishable in Crystals : 

------------------------------------------------------------------------------------

p 1

The background is not enough detailed. Why using UV and not IR ? What are the benefits which can be obtained from UV assistance ? Other people used this technics before ? with which benefits ? 

If the purpose is clearly written, the background and the why-using this technics is not enough explained. 

------------------------------------------------------------------------------------

p 2

The UV irradiation time was only of 3 h for 12 days (around 288 h) of total crystallization process. Why so little time (only 1% of the total process)  ? Different exposure time were not tried ?

------------------------------------------------------------------------------------

p 3

the crystal were analyzed by optical microscopy, no statistical analysis could be done (to obtained distribution of size of crystal) ? these would be better to show that UV has an effect on distribution of crystal size. 

For the X ray diffraction study, some Powder diffraction pattern were referenced, but without the reference litterature where they maybe were found ((PDF) 96-201-6573  and (PDF) 00-056-1522 were described in an article ?) 

------------------------------------------------------------------------------------

in conclusion the authors show clearly how the UV light irradiation change the structures of the crystal. 

With UV :

the crystal were bigger (by optical microscope) but present other form of complex (show by XRD).

This difference could be link to the fact that with UV, the UV wavelength emission is similar to the absorption of Thiourea (by UV), so the thiourea will absorb the UV and then have more energy to move. the sulfur will be nearer of the cobalt, decreasing the bond Co-Cl (by Raman). 

First I think this point (resumed p 7 line 180 to line 185) can be detailed and moved in the discussion.

Secondly the author not show enough in which extent UV light is important for their purpose. They use thiourea cobalt complex, because for example (but not only) it can be used as NLO materials. In this idea, which type of crystal are preferred ? the UV will give better crystal for NLO (only at the theoritical point of view) ? Even if the main purpose is to study the effect, one sentence only for thought can be done (not mandatory this point).

With these modifications, the article can be published in the journal Crystals. 

Author Response

(The authors gave the same response as above.)

Round 2

Reviewer 1 Report

The paper requires serious revision and additional tests before potential resubmission. See the attached file.

Author Response

Dear reviewer, the authors appreciate the valuable comments and have considered them to improve the article. Attached, in word format, the responses to the comments are sent.
